# Line-Spacing-Multiplied Optical Frequency Comb Generation Using an Electro-Optic Talbot Laser and Cross-Phase Modulation in a Fiber

Juanjuan Yan *[ID], Haiyan Dong and Yu Wang

School of Electronic Information Engineering, Beihang University, Beijing 100191, China
* Correspondence: yanjuanjuan@buaa.edu.cn

**Abstract:** An optical frequency comb (OFC) generator based on an electro-optic Talbot laser and cross-phase modulation (XPM) in a high nonlinear fiber (HNLF) is designed and demonstrated. The Talbot laser is an electro-optic frequency shifting loop that is used to produce repetition rate-multiplied pulses, and these pulses work as a pump signal that induces the XPM process in the HNLF to modulate the phase of a probe signal. At the output of the HNLF, OFCs with a multiplied line spacing can be generated. The effects of the pump power and the HNLF length on the performance of the generated OFCs are theoretically analyzed. In the experiments, the line spacing of the generated OFCs is multiplied to be 10 GHz, 15 GHz, and 20 GHz with a factor of 2, 3, and 4, respectively. The center of the OFCs is tuned in a 4 nm range by adjusting the wavelength of the probe signal.

**Keywords:** optical frequency comb; electro-optic Talbot laser; cross-phase modulation; line-spacing multiplication

## 1. Introduction

Optical frequency combs (OFCs) can be used in many applications such as arbitrary waveform generation (AWG) [1], wave division-multiplexed systems [2], frequency measurements [3], precision spectroscopy [4], etc. [5,6]. Mode-locked lasers (MLLs) were first employed for OFC generation [5,7]. However, the line spacing of generated OFC is at the order of tens of the MHz determined by the cavity length of the MLL. Some solutions have been put forward to produce OFCs with a larger spacing, including harmonic mode-locking [8], micro-resonator [9,10], electro-optic modulation of a continuous wave (CW) laser [11], and Talbot effect-based schemes [12,13]. The spacing multiplication methods based on silica microspheres [10] or Talbot effects [12,13] are more promising due to their advantage of flexibility. In particular, the latter approach has attracted more attention owing to the fact that a larger multiplication factor can be realized and that the line spacing can be multiplied from hundreds of MHz to hundreds of GHz [12]. In this method, the pulse repetition rate of a lower-rate laser source is multiplied via a temporal Talbot effect (also referred to as temporal self-imaging) by launching the pulse laser into a dispersive medium, and then the repetition rate-multiplied pulses, serving as pump signals, are co-coupled into a length of high nonlinear fiber (HNLF) with a CW probe [12,13]. In this way, the cross-phase modulation (XPM) process is induced in the HNLF, and the resulting OFC has a line spacing that is equal to the multiplied repetition rate of the pump pulses. Dispersive fibers [14] or linearly chirped fiber Bragg gratings (LCFBG) [15] can be applied to achieve a pulse repetition-rate multiplication (PRRM).

In [12], MLL pluses at a 250 MHz repetition were used and PRRM was realized through temporal Talbot effects in a dispersion compensation fiber (DCF). A multiplication factor up to 1540 was achieved, and the resulting OFC had a frequency spacing of 385 GHz. However, due to the low repetition rate of the original pulses, a long fiber was required to satisfy the Talbot condition. Also, the third-order dispersion in the DCF resulted in the distortion of

the output pulses and additional peaks around each spectral line of the generated comb [12]. To obtain a pulse train at a rate of 10 GHz, pulse compression and reshaping were applied to the generated waveform by an intensity and phase modulation of a CW light in [13]. And the rate of the pulses was then multiplied with a factor of 5 via a temporal Talbot effect in a combined dispersive fiber consisting of a 0.54 km simple single-mode fiber and a 3.02 km DCF. Similarly, the 50 GHz pulse train and a CW probe were coupled into a HNLF to induce the XPM process, and an OFC with a line spacing of 50 GHz was produced [13]. In the scheme, two relatively short dispersive fibers were used to provide a total dispersion for the realization of a pulse-rate multiplication. A propagation loss was also inevitably introduced. A LCFBG can be employed to reduce this loss [12]. However, the advances in grating fabrication, especially in terms of repeatability, are required to achieve tunable PRRM [15].

A Talbot laser is an alternative method for the pulse repetition-rate multiplication. The traditional Talbot laser is a frequency-shifted feedback laser based on an acoustic optic frequency shifter, and the working frequency is usually in the tens of MHz [16]. When an electro-optic (EO) frequency shifter is applied, a GHz Talbot laser can be achieved [17]. Here, we designed and demonstrated a line-spacing-multiplied OFC generator by a combined use of an EO Talbot laser and XPM in a HNLF, thereby avoiding the needs of MLLs and long dispersive fibers. In an EO Talbot laser, the frequency shifter is realized with a dual-parallel Mach–Zehnder modulator (DPMZM). The working principle of this OFC generator was theoretically modeled, and the influences of pump power and the length of the HNLF on the performance of the generated OFC were analyzed. In the experiments, the DPMZM was driven by a 5 GHz radio frequency (RF) signal, and the optical pulses with a repetition rate of 10 GHz, 15 GHz, and 20 GHz were, respectively, produced with the EO Talbot laser. Correspondingly, OFCs with a line spacing of 10 GHz, 15 GHz, and 20 GHz were generated after the XPM process in a 200 m HNLF. The comb line spacings were, respectively, multiplied with a factor of 2, 3, and 4. The wavelength tunability of the generated OFCs was also demonstrated in a range of 4 nm.

## 2. Principles

The schematic diagram of the proposed system for generating a line-spacing-multiplied OFC is shown in Figure 1. This OFC generator consisted of an EO Talbot laser loop and a length of HNLF. The light from a CW laser was injected into the frequency shifting loop via a 2 × 2 optical coupler (OC1), and a polarization controller (PC1) was used to adjust the polarization state of the light so as to obtain a maximum transmission efficiency. In the loop, the light was sent to a DPMZM and a carrier-suppressed single sideband (CS-SSB) was modulated by a RF signal at a frequency of $f_s$. In this way, a frequency shift of $f_s$ was achieved. The modulated laser was then amplified with an Erbium-doped fiber amplifier (EDFA1) in order to compensate for the losses in the loop, and an optical bandpass filter (OBPF1) was employed to filter out the amplified spontaneous emission (ASE) noise generated by EDFA1. A tunable delay line (TDL) was inserted before OC1 to adjust the delay time of the loop $\tau_c$.

The output of the DPMZM is described as

$$E_{DPMZM}(t) = E_{out1}(t) + E_{out2}(t)exp\left(j\frac{\pi V_{b3}}{V_{\pi DC3}}\right), \tag{1}$$

where $E_{out1}(t)$ and $E_{out2}(t)$ are, respectively, the output field of the two sub-modulators (MZM-a and MZM-b). As shown in Figure 1, the signal from the RF source was divided equally into two branches to drive the two sub-modulators with a phase difference of 90 degrees. In this case, $E_{out1}(t)$ and $E_{out2}(t)$ are expressed as

$$E_{out1}(t) = \frac{\sqrt{2}}{2}E_{in}(t)\left\{\exp\left[j\left(\frac{\pi V_{b1}}{V_{\pi DC1}} + \frac{\pi V_m \cos(2\pi f_s t)}{V_{\pi RF}}\right)\right] + \exp\left(-j\frac{\pi V_m \cos(2\pi f_s t)}{V_{\pi RF}}\right)\right\}, \tag{2}$$

$$E_{out2}(t) = \frac{\sqrt{2}}{2} E_{in}(t) \left\{ \exp\left[ j\left( \frac{\pi V_{b2}}{V_{\pi DC2}} + \frac{\pi V_m \sin(2\pi f_s t)}{V_{\pi RF}} \right) \right] + \exp\left( -j\frac{\pi V_m \sin(2\pi f_s t)}{V_{\pi RF}} \right) \right\}, \tag{3}$$

$$E_{in}(t) = \sqrt{2P_{in}} \exp(j2\pi f_0 t), \tag{4}$$

where $P_{in}$ and $f_o$ are, respectively, the average power and frequency of the input optical field to the DPMZM, $V_m$ is the amplitude of the modulation signal, $V_{\pi RF}$ is the half-wave voltage for RF signals, $V_{\pi DCx}$ ($x$ = 1, 2, 3) is the two sub-modulators and the main modulator (MZM-c) for the DC, and $V_{bx}$ is the DC bias voltage applied to the modulator. To perform CS-SSB modulation, MZM-a and MZM-b were biased at the minimum transmission point, and MZM-c was biased at the quadrature transmission point. In the case of a small signal modulation, the CS-SSB-modulated optical field output from the DPMZM is written as

$$E_{DPMZM}(t) \approx \sqrt{2P_{in}} J_1(\pi\beta) \exp[j2\pi(f_o + f_s)t], \tag{5}$$

where $J_1$ is the first-order Bessel function of the first kind and $\beta = V_m/V_{\pi RF}$ is the modulation depth. From Equation (5), it is clear that the frequency of the light was shifted by the RF frequency $f_s$ in each roundtrip. Thus, the resulting field at the output was a comb of the optical frequencies separated by $f_s$, and this is expressed as [18]

$$E_{out}(t) = \sqrt{2P_{in}} \exp(j2\pi f_o t) \times \sum_{k \geq 0} J_1^k(\pi\beta)(\sqrt{\alpha_o g})^k \exp(-j2\pi k(f_s t - f_o \tau_c)) \exp\left( j\pi k(k+1)\frac{f_s}{f_c} \right), \tag{6}$$

where $g$ represents the power gain provided by EDFA1, $\alpha_o$ is the insertion loss of the loop, and $f_c = 1/\tau_c$ is the free spectral range. When $f_s/f_c = p/q$ (where $p$ and $q$ are coprime integers), the pulse repetition rate becomes $qf_s$. In this case, a temporal Talbot effect is induced. The process results in the pulse repetition rate being multiplied to $q$ times of the original. However, the frequency spacing of the OFC expressed by Equation (6) remains $f_s$. To achieve the spacing multiplication, the effect of the XPM in the HNLF was employed here.

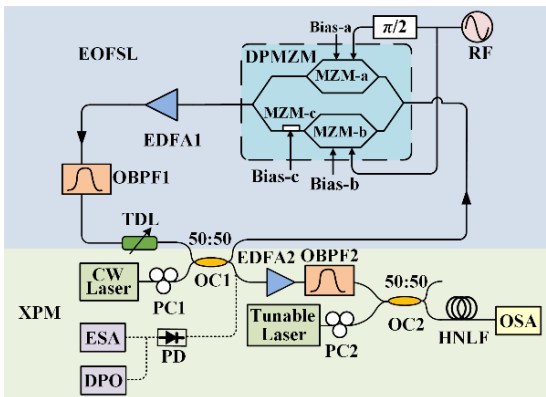

**Figure 1.** Schematic diagram of the OFC generation with a multiplied spectral line spacing. (CW laser: continuous-wave laser; PC: polarization controller; OC: optical coupler; RF: radio frequency; DPMZM: dual-parallel Mach–Zehnder modulator; EDFA: erbium-doped fiber amplifier; OBPF: optical-band pass filter; TDL: tunable delay line; HNLF: highly nonlinear fiber, PD: photo detector; ESA: electrical signal analyzer; DPO: digital phosphor oscilloscope; and OSA: optical spectrum analyzer).

As shown in Figure 1, the output pulses of the frequency shifting loop were amplified with EDFA2 and injected into another OBPF2 to filter out the ASE noise. Then, the output signals were coupled into a section of HNLF as the pump to induce the XPM effect. The probe signal was a CW from a tunable laser at a frequency of $\omega_{pb}$. After the XPM process, the optical field of the probe signal can be expressed as

$$E_{pb}(t) \propto \exp\left( j\omega_{pb}t \right) \exp[j\phi_{NL}(t)] = \exp\left( j\omega_{pb}t \right) \exp[jm_{XPM}P_{pump}(t)], \tag{7}$$

where $\varphi_{NL}(t)$ is the nonlinear phase shift introduced to the probe signal, $P_{pump}(t)$ is the instantaneous power of the pump, and $m_{XPM}$ is a complex number determined by the nonlinear coefficient, the length of the HNLF, and propagation loss, as well as the relative state of polarization between the pump and probe [19]. Here, another PC (PC2) was used to align the polarization states of the pump and the probe. In this case, when the propagation loss, dispersion, and birefringence in the HNLF were negligible, $\varphi_{NL}(t)$ was related to the pump power by [20]

$$\varphi_{NL}(t) = 2\gamma L_{HNLF} P_{pump}(t) = 2\gamma L_{HNLF} G T_f |E_{out}(t)|^2, \tag{8}$$

where $\gamma$ and $L_{HNLF}$ are the nonlinear coefficient and the length of the HNLF, respectively, $G$ is the power gain provided by EDFA2, $T_f$ is the power loss introduced by OBPF2, and $E_{out}(t)$ is the output field of the Talbot laser, which is described as Equation (6). According to Equation (7), the corresponding optical spectrum of the probe signal is written as

$$
\begin{aligned}
F_{pb}(\omega) &\propto \left| \int_{-\infty}^{+\infty} \exp\left[jm_{XPM}P_{pump}(t)\right] \exp\left[-j(\omega - \omega_{pb})t\right] dt \right|^2 \\
&\approx \left| \int_{-\infty}^{+\infty} \left[1 + m_{XPM}P_{pump}(t)\right] \exp\left[-j(\omega - \omega_{pb})t\right] dt \right|^2 = \delta(\omega - \omega_{pb}) + m_{XPM}^2 \cdot S_{RF}(\omega - \omega_{pb})
\end{aligned}
\tag{9}
$$

where the Taylor expansion of the exponential function is applied and simplified for a weaker XPM ($m_{XPM} \cdot P_{pump}(t) << 1$ radian); and $S_{RF}(\omega)$ is the RF spectrum of the pump pulses. Therefore, the optical spectrum of the probe after the XPM was proportional to the RF spectrum of the pump superimposed on the Dirac delta function, $\delta(\omega)$. If the pump was pulsed with a repetition rate of $qf_s$, the corresponding RF spectrum consisted of frequency lines spaced at $qf_s$, and the optical spectrum of the probe became an OFC with a line spacing of $qf_s$. From Equation (6), it is known that the pulse repetition rate can be multiplied to be $qf_s$ when the delay time of the frequency shifting loop is adjusted to meet the condition of $f_s/f_c = p/q$ (where $p$ and $q$ being coprime integers). So, the comb line-spacing multiplication can be achieved by tuning the TDL in the loop.

To confirm the principle described above, some simulations were performed first. In the simulations, the DPMZM was modeled as Equations (1)–(4), and the propagation of the pump pulses and the probe signal in the HNLF is described with the following nonlinear Schrödinger equations (NLSEs) [19]:

$$\frac{\partial A_p}{\partial z} + \frac{i\beta_{2p}}{2}\frac{\partial^2 A_p}{\partial T^2} + \frac{\alpha}{2}A_p = i\gamma(|A_p|^2 + 2|A_s|^2)A_p, \tag{10}$$

$$\frac{\partial A_s}{\partial z} + d\frac{\partial A_s}{\partial T} + \frac{i\beta_{2s}}{2}\frac{\partial^2 A_s}{\partial T^2} + \frac{\alpha}{2}A_s = i\gamma(|A_s|^2 + 2|A_p|^2)A_s \tag{11}$$

where the subscripts of $p$ and $s$ denote the pump pulses and the probe signal, respectively, $A_y$ ($y$ is $p$ or $s$) is the slowly varying envelop of the laser in the HNLF at the position $z$, $T = t - z/v_{gy}$ is a reference time, $v_{gy}$ is the group velocity, $d = (v_{gp} - v_{gs})/v_{gp}v_{gs}$ is the walk-off parameter, $\beta_{2y}$ is the group-velocity dispersion (GVD) parameter of the fiber, and $\alpha$ is the attenuation coefficient. So, the effects of GVD, self-phase modulation (SPM), and XPM in the HNLF were taken into account.

In our study, the frequency of the RF signal was $f_s$ = 5 GHz. The wavelength of the pump laser and the probe signal was 1550 nm and 1553.1 nm, respectively. The dispersion coefficient and dispersion slope of the HNLF were −2.569 ps/nm/km and 0.005 ps/nm$^2$/km @1550 nm, respectively. The attenuation coefficient and nonlinear coefficient were 1.5 dB/km and 11 W$^{-1}$km$^{-1}$, respectively. The parameter values used in the simulations were the same as those applied in the experiments. The propagation process of the pump and the probe in the HNLF was simulated by numerically solving the NLSEs using the split-step Fourier method [19]. With $p$ = 1 and $q$ = 2, 3, 4, respectively,

the pump pluses from the EO Talbot laser were obtained, as presented in Figure 2a. It is clear that the pulse repetition rate was, respectively, 10 GHz, 15 GHz, and 20 GHz, i.e., the PRRM with a factor of 2, 3, and 4 was realized. The corresponding RF power spectra of these pulses are shown in Figure 2b. When the length of the HNLF was $L_{HNLF}$ = 200 m and the average pump power injected into the HNLF was $P_{av\_pump}$ = 14 dBm, it resulted in OFCs that were centered at the probe wavelength, as shown in Figure 2c. By comparing Figure 2b,c, it can be found that the line spacing of the two spectra for a given $q$ was equal to $qf_s$, and the envelope of an OFC spectrum was approximately the superposition of a RF spectrum envelope and the probe spectral line, as concluded from Equation (9).

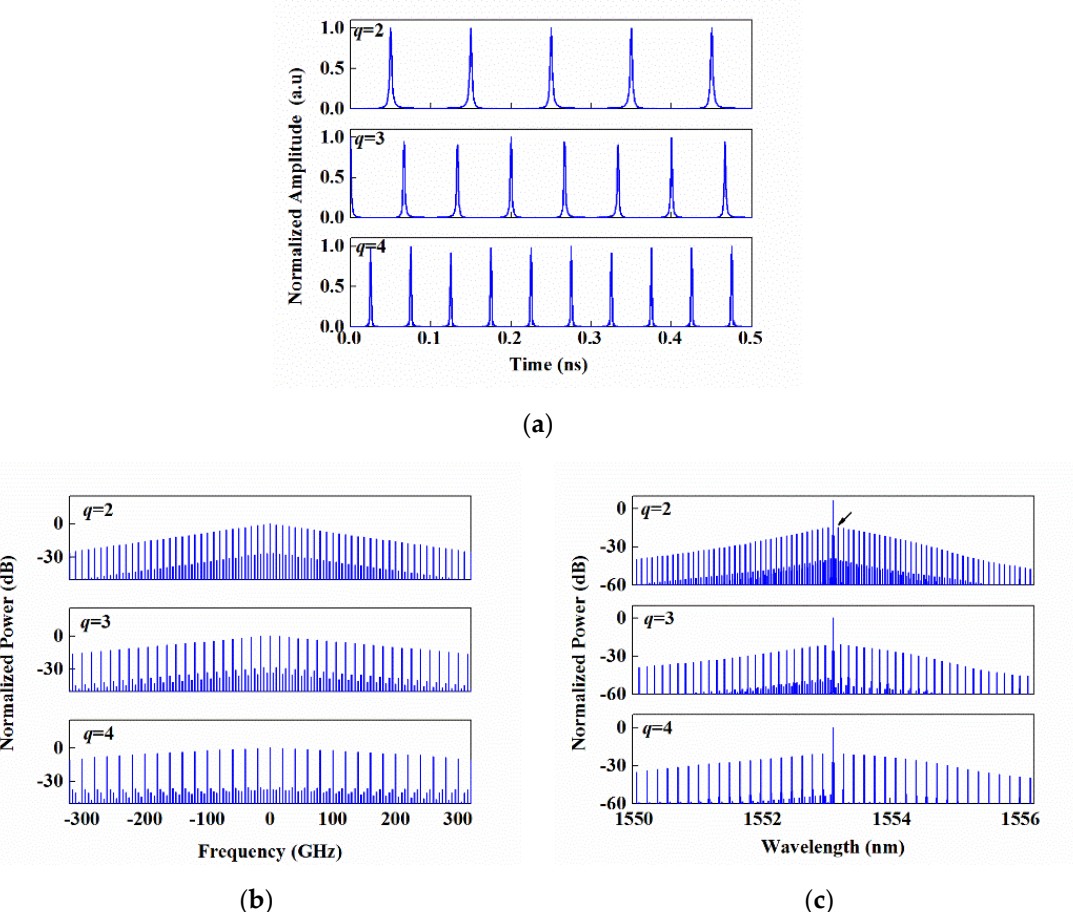

**Figure 2.** The simulated results, respectively, with $p$ = 1 and $q$ = 2, 3, 4, (**a**) the pump pulses output from the EO Talbot laser, (**b**) the RF power spectra of the pulses in (**a**,**c**), and the produced OFCs at the output of the HNLF.

The right comb line nearest to the probe spectral line, which is pointed to by the arrow in Figure 2c, with $q$ = 2 was taken as an example to investigate the influences of the HNLF parameters on the performance of the generated OFCs, and the side-mode suppression ratio (SMSR) and peak power of this line were also taken into account. The results are shown in Figure 3a with $L_{HNLF}$ = 200 m. It can be seen that the SMSR increased and then decreased with the increase in the pump power while the peak of the line kept increasing in these considered cases. When $P_{av\_pump}$ was set to 6 dBm, the SMSR had a peak value of about 26.5 dB. In this case, the line-spacing-doubled OFC with $P_{av\_pump}$ = 14 dBm and 22 dBm were also shown as a comparison, as presented in Figure 3b. From Figure 3b, it is also clear that more lines were generated in the range of 6 nm due to the fact that the higher the pump power, the stronger the XPM nonlinearity. However, the SMSR of each line decreased for the simultaneously enhanced four-wave mixing (FWM) interactions among all the modes [19]. Also, Equation (9) was not applicable since the nonlinear phase induced

by XPM became larger. For the case of $P_{av\_pump}$ = 22 dBm in Figure 3b, the nonlinear phase was $\varphi_{NL} = 2\gamma L_{HNLF}P_{av\_pump} \approx 0.7$ radian. Consequently, the envelop of the resulting OFC could not be regarded as a superposition of the RF spectrum of the pump and the probe spectral line. As a compromise, the pump power $P_{av\_pump}$ was set to be 14 dBm in the following.

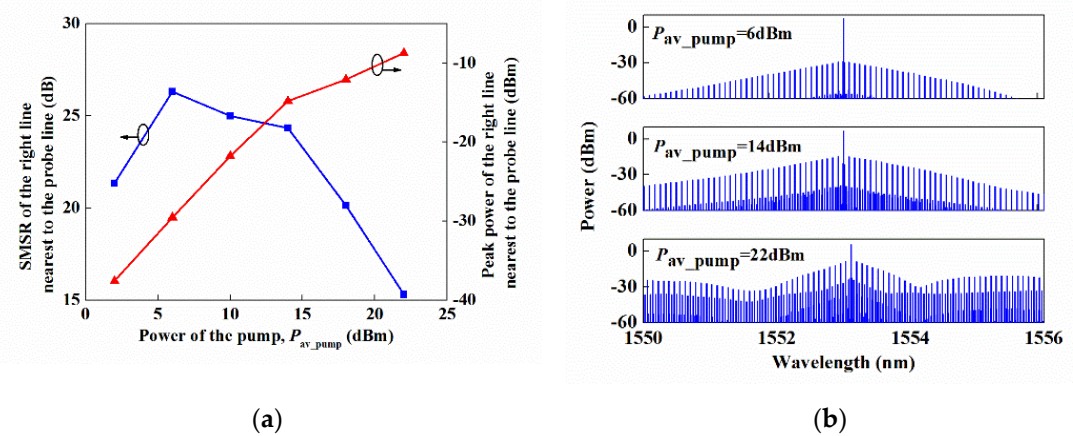

(**a**)                                                           (**b**)

**Figure 3.** The simulated results with $p$ = 1 and $q$ = 2, (**a**) the SMSR and the peak power of the right line nearest to the probe line as a function of the pump power injected into the HNLF, and (**b**) the generated OFCs, respectively, with a $P_{av\_pump}$ = 6 dBm, 14 dBm, and 22 dBm.

Similarly, the SMSR and peak power of the considered line are, respectively, shown in Figure 4a as a function of the HNLF length $L_{HNLF}$. It could be observed that the SMSR also increased and then decreased with $L_{HNLF}$ increasing while the peak of the line kept increasing in all of the considered cases. When the $L_{HNLF}$ was 100 m, the SMSR had a peak value of about 26 dB. Figure 4b shows the resulting OFCs with an $L_{HNLF}$ = 100 m, 200 m, and 600 m. It could also be found that more comb lines were generated with the length of the HNLF, which increased since the XPM nonlinearity was accumulated along the HNLF. However, the SMSR decreased with an $L_{HNLF}$ > 100 m, which was also mainly due to the enhanced FWM efficiency. As a compromise of the SMSR and the bandwidth, a 200 m HNLF was applied in our experiments.

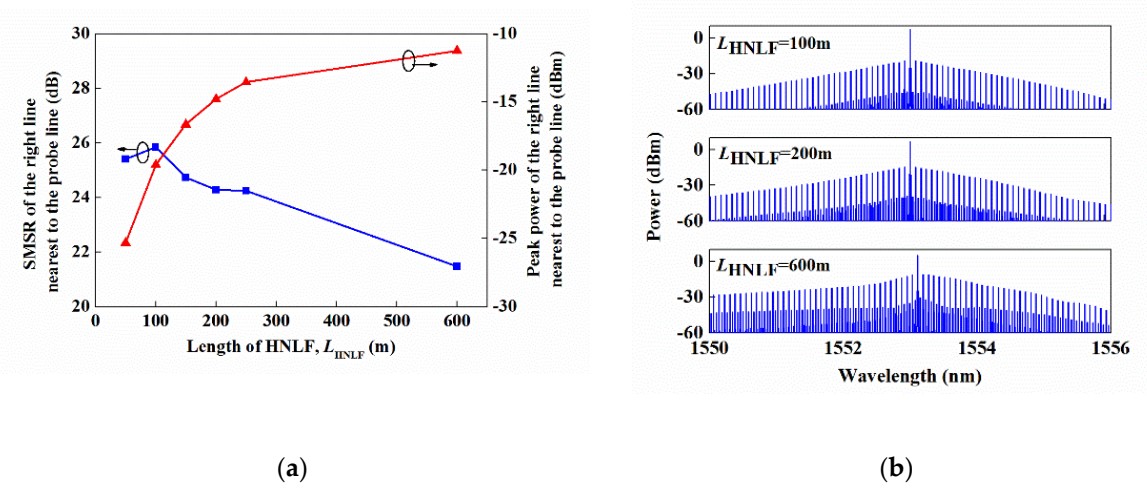

(**a**)                                                           (**b**)

**Figure 4.** The simulated results with $p$ = 1 and $q$ = 2, (**a**) the SMSR and the peak power of the right line nearest to the probe line as a function of the HNLF length, and (**b**) the generated OFCs, respectively, with an $L_{HNLF}$ = 100 m, 200 m, and 600 m.

## 3. Experimental Results

Proving experiments on the system were also performed, as shown in Figure 1. A CW laser (RIO Orion laser module) working at 1549.76 nm was employed, and the output power was 9 dBm. The used DPMZM (Thorlabs LN86S-FC) had a bandwidth of 14 GHz, and the half-wave voltages of the two sub-MZMs and the main modulator were 4.5 V and 6 V, respectively. The modulator was driven by a 5 GHz RF signal to perform a CS-SSB modulation. A home-built EDFA1 was applied to provide an optical gain of 18 dB. The bandwidth of the OBPF1 was about 7 nm with a center of 1550 nm. The tunable range of the used TDL (GP-UM-MDL-002) was 0~330 ps. The length of the EO Talbot laser loop was about 25.5 m, and the roundtrip delay was calculated to be about 124.1 ns with the refractive index of the fiber core being equal to 1.46. The optical pulses after frequency multiplication were detected by a PD and observed with a digital phosphor oscilloscope (DPO, Tektronix DPO72004). The responsivity of the PD was 0.55 A/W and the bandwidth was 30 GHz. The pump pulses from the Talbot laser were amplified and then passed through an OBPF2 (Finisar WaveShaper 1000S) with a bandwidth of 1.6 nm. The probe light was from a tunable CW laser (Optilab TWL-C-M) and the power was 6 dBm. The length of the employed HNLF (YOFC, NL1550-NEG) was 200 m. The dispersion coefficient and dispersion slope of the fiber were the same as those used in the simulations. The nonlinear coefficient was measured as 10.7 $W^{-1}km^{-1}$, with the method based on a measurement of the self-phase modulation effect that induced the nonlinear phase shift [21]. The optical spectra obtained in the experiments were monitored by the OSA (YOKOGAWA AQ6370C).

Figure 5a shows the output spectrum of the DPMZM when the loop was not closed, and it was clear that CS-SSB modulation was realized with a −1st-order sideband remaining. The side-mode suppression ratio was measured as 18.5 dB. When the loop was closed, an OFC was generated, as presented in Figure 5b. As expected, the newly generated spectral lines lay in the region of a longer wavelength. The net gain of the loop was less than the unity required to obtain a stable spectrum [22]. As a result, the intensity of the newly generated line gradually decreased. The line spacing of the OFC was 0.04 nm, which corresponded to 5 GHz at a wavelength of 1550 nm. At the same time, the period of the generated pulses was 0.2 ns, as shown in Figure 5c.

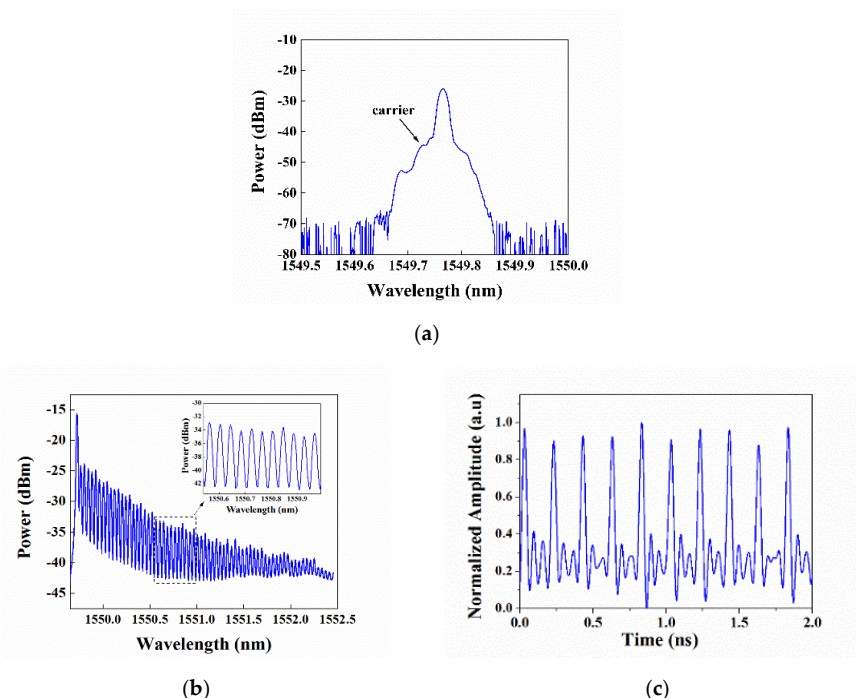

(**a**)

(**b**)          (**c**)

**Figure 5.** Experiments results, (**a**) the output spectrum of the DPMZM with a CS-SSB modulation being achieved, (**b**) the generated OFC, and (**c**) pules at the output of the loop.

In the experiments, the TDL was adjusted to change the ratio of $f_s/f_c$ so as to induce temporal Talbot effects. When the time delay was set to 0 ps, 34 ps, and 51 ps, the corresponding Talbot effects with $f_s/f_c = p/q = 1241/2, 1862/3$, and $2483/4$ were realized, respectively. The output pulses are shown in Figure 6a. The repetition frequencies of the pulses were 10 GHz, 15 GHz, and 20 GHz, i.e., they were 2 times, 3 times, and 4 times of the repetition rate of the original pulses, respectively. It was clear that the amplitude fluctuations became more obvious with the multiplication factor increasing. Some facts lead to this phenomenon. Firstly, the fluctuations mainly resulted from the noise introduced by the EDFA1, the laser, the RF source, and the PD. Secondly, the bandwidth of the generated OFC was limited, as shown in Figure 5b. Consequently, the pulses were not narrow enough, which resulted in the pulse overlapping with the increase in the multiplication factor. Finally, the observed results were also affected by the limited 20 GHz bandwidth of our used DPO. The corresponding RF power spectra of these pulses are presented in Figure 6b. It can be seen that the 5 GHz and 15 GHz tones were suppressed with a ratio pf about 20 dB against other harmonics of 10 GHz and 20 GHz with $f_s/f_c = p/q = 1241/2$, thereby indicating that a pulse repetition multiplication of 2 was well achieved. However, the harmonics suppression ratio decreased for the other two cases, which also revealed a degraded pulse rate multiplication, as shown in Figure 6b. This could be improved by optimizing the power and the net gain in the loop so as to increase the bandwidth of the generated OFC. Here, the OFC spectrum output from the EO Talbot laser was unchanged while the pulse rate was multiplied. To multiply the comb line spacing, after further amplifying and filtering the pump pulses, they were coupled into the HNLF with a 1553.1 nm probe CW from the tunable laser. When the pump power injected into the HNLF was 14 dBm, the corresponding probe spectra at the output of the HNLF were obtained, as shown in Figure 6c. It can be found that the line spacing of the OFC was, respectively, multiplied to be 0.08 nm, 0.12 nm, and 0.16 nm, i.e., 10 GHz, 15 GHz, and 20 GHz when the pump pulse rate multiplication factor was 2, 3 and 4, respectively. So, owing to the XPM process, the OFCs with line spacing multiplied by a factor of 2, 3, and 4 were generated.

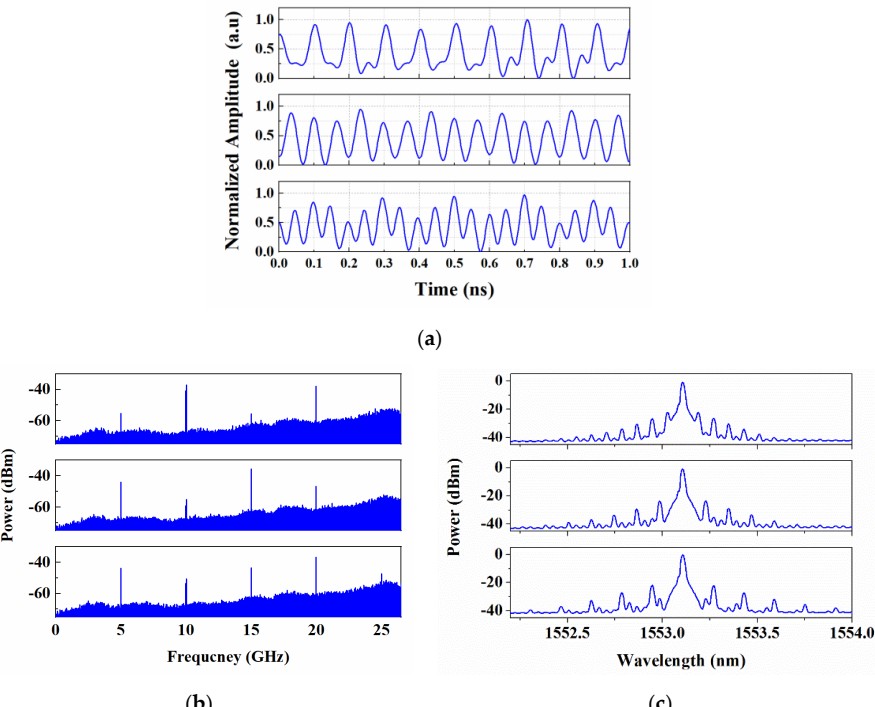

**Figure 6.** Experimental results with, respectively, $p/q = 1241/2, 1862/3$, and $2483/4$, (**a**) the generated pump pulses from the EO Talbot laser, (**b**) the corresponding RF power spectra of the pulses in (**a**,**c**), and the spectra of the probe light at the output of the HNLF.

As shown in Figure 6c, the center of the output OFC was determined by the working wavelength of the probe light. So, this scheme was also wavelength-tunable. In our experiments, the wavelength of the probe signal was tuned from 1553 nm to 1557 nm. The generated OFCs, respectively, at the center wavelength of 1553.1 nm, 1554.7 nm, and 1556.3 nm are presented in Figure 7 with a $f_s/f_c = p/q = 1241/2$ and a pump power of 22 dBm. From this figure, it can be seen that the OFC spectral envelope shapes centered at different wavelengths were similar. Compared with the spectrum in Figure 6c, the power of each frequency line increased since the XPM process was enhanced by the increased pump power. And the spectra became asymmetrical around the center wavelength, which resulted from the chirps of the pump pulses. This conclusion can also be drawn from Figure 3b with a $P_{av\_pump}$ = 22 dBm.

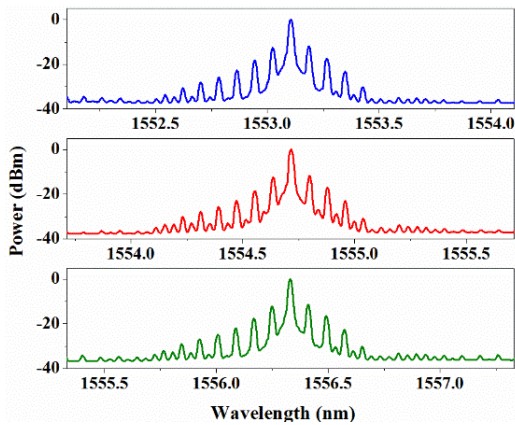

**Figure 7.** The probe spectra at the output of the HNLF with $f_s/f_c = p/q = 1241/2$, as well as the CW probe at the respective wavelengths of 1553.1 nm, 1554.7 nm, and 1556.3 nm.

## 4. Discussion

From the above theoretical and experimental results, it is clear that OFC line-spacing multiplication can be achieved using our scheme. The multiplication factor was tuned by adjusting the TDL in the EO Talbot laser loop. Compared with the existing methods in [12,13], our method avoids the use of MLLs and long dispersive fibers. The bandwidth of the generated OFC was also determined by the bandwidth and power of the pump from the laser loop except for the length and the nonlinear coefficient of the HNLF. More comb lines can be experimentally generated in the Talbot laser by increasing the net gain and optimizing the power in the loop, which contributes to generating pump pulses with a narrow width and a larger peak power. In this way, the line-spacing multiplication factor and the bandwidth of the resulting OFC could be increased.

Finally, it should be pointed out that the product of the nonlinear coefficient and the length of the used HNLF also need to be optimized to further improve the SMSR performance of the line-spacing-multiplied OFC for the FWM effect that also exists in the HNLF. A section of the HNLF with a small dispersion was required to decrease the FWM efficiency.

## 5. Conclusions

A scheme for the generation of line-spacing-multiplied OFCs was proposed and demonstrated. This OFC generator is composed of an EO Talbot laser and a length of HNLF. The frequency shifter in the Talbot laser is a DPMZM that performs CS-SSB modulation. The pulses output from the Talbot laser work as pumps to stimulate the XPM process in the HNLF. By adjusting the TDL in the Talbot laser loop, the repetition rate of the pump pulses can be multiplied, and the line spacing of the generated OFCs at the output of the HNLF can be correspondingly multiplied with the same factor. The principle of this OFC generator was theoretically modeled and simulated to investigate the influences of the pump power and the HNLF length on the performance of the generated OFCs. In the

experiments, the DPMZM was driven by a 5 GHz RF signal. OFCs with a line spacing of 10 GHz, 15 GHz, and 20 GHz were produced with a multiplication factor of 2, 3, and 4, respectively. If the optical power and the net gain in the Talbot laser loop is optimized, the bandwidth of the generated OFC can be increased, and a greater multiplication factor can be realized. The centers of the generated OFCs were determined by the wavelength of the probe light, and they can be tuned while keeping a similar spectral envelop. So, this scheme has the advantage of wavelength tunability with no need for MLLs and long dispersive fibers. It can be applied in the optical arbitrary waveform generation and wave division-multiplexed systems.

**Author Contributions:** Conceptualization, J.Y.; methodology, J.Y.; software, H.D. and Y.W.; validation, J.Y.; investigation, H.D. and J.Y.; data curation, H.D. and Y.W.; writing—original draft preparation, H.D. and J.Y.; writing—review and editing, J.Y.; visualization, H.D.; supervision, J.Y.; funding acquisition, J.Y. All authors have read and agreed to the published version of the manuscript.

**Funding:** This research was supported by the National Natural Science Foundation of China (NSFC) (grant numbers: 61771029 and 61201155).

**Institutional Review Board Statement:** Not applicable.

**Informed Consent Statement:** Not applicable.

**Data Availability Statement:** The data presented in this study are available on request from the corresponding author due to privacy.

**Conflicts of Interest:** The authors declare no conflicts of interest.

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
