# Peer review of "Line-Spacing-Multiplied Optical Frequency Comb Generation Using an Electro-Optic Talbot Laser and Cross-Phase Modulation in a Fiber"

_photonics, doi:10.3390/photonics11030282_

Round 1

Reviewer 1 Report

Comments and Suggestions for Authors

The authors proposed and demonstrated a fiber scheme for the generation of line-spacing multiplied optical frequency combs (OFCs). This OFCs generator is composed of an EO Talbot laser and highly nonlinear fiber (HNLF). The principle of this OFCs generator is explained and numerically simulated to investigate the performance of the generated OFCs depending on parameters. OFCs with a line spacing of 10 GHz, 15 GHz and 20 GHz are respectively produced with a multiplication factor of 2, 3 and 4. The results are useful and interesting. I have only a few comments and suggestions for the authors.

1. It would be useful if the authors say a little more about HNLF. Is it commercial or custom made fiber? Lines 145-146 give its characteristics. Please note that in lines 217-218 the same parameters are repeated. It is written that the nonlinear coefficient was measured experimentally. What about dispersion? It should be mentioned whether these are experimental values or calculated ones.

2. Lines 146-147. “The nonlinear process of XPM in the HNLF is simulated by numerically solving the nonlinear Schrödinger equations using the split-step Fourier method [15].” What effects were taken into account in the nonlinear Schrödinger equations? It would be useful to write the master equation. Does Raman and/or Rayleigh scattering affect the process?

3. In the introduction, microresonator OFCs are mentioned. Note that line-spacing multiplied OFCs in microresonators were demonstrated in [https://doi.org/10.1364/OL.34.000878; https://doi.org/10.1364/OL.44.003078; https://10.1109/LPT.2021.3068373] and this fact should be added.

4. There are typos and inaccuracies in the text. For example:
Line 144 contains a Chinese character.
In the caption to Fig. 6, “(c)” is missed.

Reviewer 2 Report

Comments and Suggestions for Authors

The paper shows  the development of a frequency comb exploiting the cross phase modulation effect of an electro-optic Talbot laser in a high nonlinear fiber. Even if a GHz Talbot laser using electro-optic frequency shifter is not a novelty I think that the added ingredients are innovative and I agree for pubblication with following minor revisions.

- I note few typos like "if the pump is pulses" instead of "if the pump is pulsed" or Thorlab instead of Thorlabs

- I suggest to explain more clearly how the eq 3 is obtained

- The references on frequency comb do not cite papers of nobel prize Theodor W. Hansch comb and I think that is fair given its merits in the comb development and diffusion.

- I do not see references pointing out the importance of the frequency comb in precision spectroscopy like [Physical Chemistry Chemical Physics 18 (25), 16715-16720]

-I suggest to introduce a reference for equation 1 or at least use exponential (more widespread) instead of Sin and Cos notation 
